# Antioxidant and Anti-Inflammatory Effects of Coenzyme Q10 Supplementation on Infectious Diseases

**DOI:** 10.3390/healthcare10030487

**Published:** 2022-03-07

**Authors:** Sonia Sifuentes-Franco, Dellaneira Carolina Sánchez-Macías, Sandra Carrillo-Ibarra, Juan José Rivera-Valdés, Laura Y. Zuñiga, Virginia Aleyda Sánchez-López

**Affiliations:** 1Laboratory of Biological Systems, Department of Health Sciences, Valles Campus (CUValles), University of Guadalajara (UDG), Ameca 46600, Jalisco, Mexico; 2Department of Health Sciences—Disease as an Individual Process, Tonalá Campus, University of Guadalajara (UDG), Tonalá 45425, Jalisco, Mexico; lauray.zuniga@academicos.udg.mx; 3School of Health Sciences, Universidad del Valle de México, Zapopan 45010, Jalisco, Mexico; macias10caro@gmail.com (D.C.S.-M.); sann_dy31@yahoo.com.mx (S.C.-I.); 4Institute of Translational Nutrigenetics and Nutrigenomics, Department of Molecular Biology and Genomics, Health Sciences Campus (CUCS), University of Guadalajara (UDG), Guadalajara 44100, Jalisco, Mexico; juanjoseon@gmail.com; 5Pharmacovigilance Department, Laboratorios PISA, Guadalajara 44100, Jalisco, Mexico; aleyda_sanmis7@hotmail.com

**Keywords:** CoQ10, ubiquinone, infectious disease, antioxidant, inflammation and immunomodulator

## Abstract

With the appearance of new viruses and infectious diseases (ID) such as COVID-19 in 2019, as well as the lack of specific pharmacological tools for the management of patients with severe complications or comorbidities, it is important to search for adjuvant treatments that help improve the prognosis of infectious disease patients. It is also important that these treatments limit the oxidative and hyperinflammatory damage caused as a response to pathogenic agents, since, in some cases, an inflammatory syndrome may develop that worsens the patient’s prognosis. The potential benefits of complementary nutrients and dietary interventions in the treatment of pathological processes in which oxidative stress and inflammation play a fundamental role have been widely evaluated. Coenzyme Q10 (CoQ10) is a supplement that has been shown to protect cells and be effective in cardiovascular diseases and obesity. Additionally, some studies have proposed it as a possible adjuvant treatment in viral infections. Preclinical and clinical studies have shown that CoQ10 has anti-inflammatory and antioxidant effects, and effects on mitochondrial dysfunction, which have been linked to the inflammatory response.

## 1. Introduction

In 1957, a substance was discovered with a structure containing a benzoquinone ring attached to a chain of 10 isoprene units: coenzyme Q10 (CoQ10) [1]. Coenzyme Q10 (CoQ10) is an essential compound in the human body that is synthesized in the inner mitochondrial membrane [1], with a lipophilic character that makes it easy to diffuse through membranes [2]. Like other compounds, CoQ10 has been considered as a possible candidate in the adjunctive treatment of chronic diseases in which oxidative stress (OS) plays an important role in the pathophysiology [3]. The search for nutrients or compounds with antioxidant, immunomodulatory and anti-inflammatory capabilities and evidence for their properties is of great interest in the scientific community, especially for diseases where an unhealthy diet influences their prevention, as is the case for infectious diseases (ID) [4]. ID have become more prevalent in recent decades; causes of this increase include environmental changes due to human activity, increased international mobility, and microbial adaptations and resistance [5]. ID are generally caused by microorganisms. Their clinical importance depends on the damage they cause to the host. Damage to tissues and organs after infection results mainly from the growth and metabolic processes of infectious agents intracellularly or within body fluids, with the production and release of toxins or enzymes that interfere with normal organ functions and/or or systems. Endogenous mechanisms can also contribute to increased damage, including the exacerbated inflammatory response of the host [6]. Depending on the causal agent of disease, the ideal treatment comprises antimicrobials, a large group of compounds with diverse structures and mechanisms of action that interfere with the growth of bacteria, viruses, fungi and parasites. Depending on their use, they are called antibiotics, antivirals, antifungals, antimycotics or antiparasitics [5]. Unfortunately, the number of antimicrobial agents in clinical development is woefully insufficient to keep pace with growing antibiotic resistance [7]. In addition to the use of antimicrobial drugs for the treatment of infectious diseases, nutritional status and diet modulate processes such as inflammation, immune function and oxidative status, which can favour the patient’s prognosis [4,8]. Therefore, the objective of this review is to present scientific evidence from clinical data that demonstrates the beneficial effects of CoQ10 on ID, evaluating its immunomodulatory, anti-inflammatory and antioxidant effects.

## 2. Data Source

An exhaustive compilation of the relevant literature on the effects of coenzyme Q10 supplementation in infectious diseases was carried out through an electronic search of the PubMed (www.pubmed.ncbi.nlm.nih.gov/, accessed on 8 January 2022), Web of Science (www.webofknowledge.com, accessed on 15 January 2022) and Scopus (www.scopus.com, accessed on 20 January 2022) databases to retrieve studies published up to December 2021. The following search criteria was used: (CoQ10 OR Ubiquinone OR ubiquinol) AND (viral infections OR infectious disease OR bacterial infections) AND (Antioxidants OR Anti-inflammatory effects). Additionally, other relevant references of identified studies were retrieved by cross-references. The research had no geographic area or language restrictions.

## 3. Coenzyme Q10

CoQ10 is a fundamental element in metabolic processes such as mitochondrial oxidative phosphorylation. It is also considered a powerful fat-soluble antioxidant. CoQ10 is present in all cell membranes and is biosynthesized in all tissues from its precursor, the 4-hydroxybenzoate ring (4HB), a derivative of tyrosine. The isoprene chain is synthesized through the mevalonate pathway, a common pathway for cholesterol synthesis [9,10]. CoQ10 (also known as ubiquinone, because it is ubiquitous) exists in its oxidized (ubiquinone) or reduced form (ubiquinol), the latter being the most predominant in the human body [11]. Although CoQ10 has similar biological functions to vitamins, it differs in that it is biosynthesized in the body and vitamins are obtained only from the diet. About half of the body’s CoQ10 is obtained through the diet and the rest is biosynthesized endogenously; major dietary sources of CoQ10 include meats, fish, salmon, sardines, pork, chicken, nuts, soybeans, vegetable oils, there are other sources that also contain CoQ10 although at much lower levels, dairy products, vegetables, fruits and cereals [12]. The daily requirement of CoQ10, both from endogenous biosynthesis and from dietary sources, is approximately 500 mg [13]. There are no established minimum or maximum effective doses; consumption of 200 mg twice a day is required to achieve a therapeutic blood level of >2.5 µg/mL, although the doses used have varied according to the type of disease [14]. Regarding its modulatory effect on inflammation, supplementation with 100–300 mg/day for 2–3 months reduced pro-inflammatory parameters [15]. There are several processes in which OS plays a fundamental part, including as aging, neuro- 69 degenerative illnesses, diabetes, and autoimmune disorders, where a decreasing level of CoQ10 is evident, and supplementation with CoQ10 has considerable health advantages [16,17,18,19,20].

Experimental data from preclinical and clinical safety studies have shown that CoQ10 does not cause adverse effects in humans and is safe for use as a dietary supplement. Indeed, acute toxicity studies in rodents have shown that the maximum tolerated CoQ10 doses are >4000 mg/kg. In a subacute toxicity study of CoQ10 in rats, doses of 0, 40, 200 or 1000 mg/kg/day were administered orally for 5 weeks, with no signs of toxicity. Chronic toxicity studies have also shown that CoQ10 administration is quite safe: none of the doses produced signs of toxicity based on haematological analysis, biochemical tests on blood or urine samples and post-mortem examinations. CoQ10 was well tolerated at doses up to 600 mg/kg/day by male and female rats [21].

## 4. Mechanism of Coenzyme Q10 Action Associated Antioxidant and Anti-Inflammatory Activity

During the infectious process, there could be an imbalance between oxidative and antioxidant species. As part of the body’s response to attack pathogens, the expression of myeloperoxidase, NADPH oxidase and nitric oxide synthase (iNOS) is induced in phagocytic cells, leading to increased production of ROS and RNS. In infection there is abundant iNOS activity in macrophages and leukocytes leading to increased production of nitric oxide (NO), which in the presence of superoxide reacts to form highly toxic peroxynitrite. The oxidants generated activate the redox-sensitive NF-κB signalling pathway, which promotes the expression of proinflammatory cytokines, chemokines and cell adhesion receptors, all of which are involved in the radical production and persistence of inflammation [22].

### 4.1. Antioxidant Functions

The most important function of CoQ10 is as a carrier in the mitochondrial electron transport chain. CoQ10 is responsible for the transport of electrons from complex I (NADH dehydrogenase) to complex II (succinate dehydrogenase); up to this point, CoQ10 is in a permanent balance between a reduced form after receiving two electrons: Ubiquinol, and an oxidized form, Ubiquinone. In the respiratory chain, this redox cycle occurs by a two-step transfer of one electron each, producing the semiquinone intermediate [20]. Likewise, CoQ10 transfers protons to the mitochondrial intermembrane space, favouring the formation of a proton gradient capable of generating the energy to produce adenosine triphosphate (ATP); this means that it has important functions in the supply of cellular energy [23,24].

Due to these electron transport properties, its function as an antioxidant is of great relevance. It has been shown to protect cell membranes from lipoperoxidation caused by free radicals (FR), and being present in all tissues, its function is very important [15]. It is assumed that the reduced form of CoQ10, ubiquinol, is the active antioxidant agent involved in most of its functions. CoQ10 is converted to reduced form of coenzyme Q10 (CoQ10H2) with electrons provided by other redox reactions in the mitochondria, it is reduced by mitochondrial glycerol 3-phosphate dehydrogenase, mitochondrial dihydroorotate dehydrogenase, and electron-carrying flavoprotein dehydrogenases (ETFDHs) [24]. For this reason, the CoQ10 redox cycle takes place in cell membranes, in which different oxidoreductases such as cytochrome b5 reductase (CytB5) and NAD(P)H quinone dehydrogenase 1 (NQO1) participate. This enables the recovery of the active form of CoQ10, and in turn, ubiquinol interacts with α-tocopherol and vitamin C radicals, decreasing OS [25]. Reducing compounds such as ubiquinol can regenerate alpha-tocopherol from the tocopherol radical. This recycling is dependent on the activity of mitochondrial succinate dehydrogenase, which provides the reducing equivalents for the conversion of ubiquinone to ubiquinol. Microsomal NADPH-dependent reduction of the phenoxyl radical in vitamin E achieves maximum efficiency in the presence of CoQ10 [26].

### 4.2. Anti-Inflammatory and Immune Functions

The antioxidant properties of CoQ10 can protect all cells and tissues, especially cells involved in the innate and adaptive immune response. OS play an important role in immunological cytotoxicity against pathogens, through the production of reactive oxygen species (ROS) by macrophages [27,28]. Research suggests that CoQ10 mediates its beneficial effects through direct and indirect anti-inflammatory mechanisms; CoQ10 has been reported to regulate the gene expression of Interleukin (IL) 1 IL-1 and tumour necrosis factor α (TNF-α) in patients with diabetic nephropathy who received supplementation [29]. There are also studies that support the idea that CoQ10 exerts anti-apoptotic and anti-inflammatory activities, through redox-dependent mechanisms, since it has been shown that supplementation with CoQ10 decreases plasma levels of C-Reactive Protein (CRP), IL-6 and TNF-α [30]. Although the specific mechanism of action of CoQ10 as an anti-inflammatory is not clear, several potential mechanisms could explain it. CoQ10 could play a role in reducing the production of pro-inflammatory cytokines by inhibiting the expression of the gene encoding nuclear factor kappa B (NF-Κb), lipoperoxides present in pathogens and oxidizing agents generated during the infectious process induce the signalling pathway in monocytes that activates the NF-kB factor, resulting in the release of TNF-alpha being favoured. The antioxidant power of CoQ10 can reduce the signalling pathway due to its radical scavenging activity [31]. Other studies suggest that the anti-inflammatory effect of CoQ10 may be associated with adiponectin; by causing an increase in CoQ10 levels, supplementation leads to a rise in adiponectin levels, which then leads to a decrease in the inflammatory response mediated by TNF-α [32]. A randomized placebo-controlled study in which selenium supplementation combined with CoQ10 was administered to healthy elderly subjects with low selenium levels showed that this supplementation reduces the inflammatory response through a decrease in plasma CRP levels [33]. Likewise, another study also evaluated the effect of administration of CoQ10 in elderly subjects on inflammatory response markers such as osteopontin, osteoprotegerin, tumour necrosis factor receptor (TNFr) 1 (TNFr1) and TNFr2, finding a reduction in these biomarkers in patients treated with selenium and CoQ10 compared to those receiving placebo [34].

Another pathology related to oxidative stress and inflammation is cancer. Although there are few studies on CoQ10 supplementation as an adjuvant in cancer treatment, there is evidence of remission after supplementation with this compound. In addition, CoQ10 has demonstrated anti-inflammatory effects and decreased oxidative stress in patients with hepatocellular carcinoma [35,36,37]. It is believed that the anti-inflammatory actions of CoQ10 in cancer may be mediated by inhibition of the activation of NF-κB transcription factors. Likewise, studies in patients with migraine have reported that CoQ10 supplementation at a dose of 400 mg/day for three months significantly reduces the levels of calcitonin gene-related peptide (CGRP) and TNF-α [38].

Most of the available information on the anti-inflammatory effects of CoQ10 regards metabolic diseases such as diabetes, obesity and hypertension. However, many other pathophysiological processes such as infections and the immune response to pathogens also present with redox imbalance. It has been shown that, in patients with endogenous antioxidant deficiency, the prognosis can be serious, so the use of CoQ10 could be beneficial. Given the recent appearance of new viruses such as Severe Acute Respiratory Syndrome Coronavirus 2 (SARS-CoV-2) and the daily increase in cases, as well as the high mortality in some patients with risk factors or previous comorbidities, the scientific population is seeking new pharmacological strategies to support a better prognosis for these patients (Figure 1).

## 5. CoQ10 as a Possible Adjuvant in Infectious Diseases

As in cancer, migraine and aging, the role of inflammation in infections, both viral and bacterial, is well known. Likewise, it is known that oxidative stress plays a fundamental role in the immune response against pathogens [39]. For this reason, as a result, it is reasonable to believe that the use of substances with antioxidant and immunomodulatory effects can serve as an adjuvant in the management of infectious diseases. There is evidence that, in patients with infections such as influenza [40] or COVID-19 [41], endogenous CoQ10 levels are significantly lower than in healthy controls, as well as showing significant correlation with various inflammatory biomarkers. A computational study showed that CoQ10, like other quinones, exhibits a possible mechanism of action consistent with the anti-inflammatory drugs methylprednisolone and embelin [42], compounds that have been shown to inhibit viral infection through a variety of mechanisms and is active against influenza virus and hepatitis B [43].

A number of studies that have evaluated the effect of CoQ10 supplementation in infectious diseases (Table 1). In a study by Soltani et al., patients received 200 mg of CoQ10 daily for 7 days, resulting in a decrease in the levels of IL-6, TNF-α, glutathione peroxidase and malondialdehyde (MDA) [44].

Another similar study evaluated the effect of the administration of ubiquinol in patients with severe sepsis and septic shock; the study yielded inconclusive results on inflammatory markers and endothelial dysfunction, as there were no significant changes after treatment in levels of TNF-α, cytochrome C, nuclear and mitochondrial DNA [45]. The limitations of this study include the short follow-up time, which could have led to no significant changes being observable, as well as the severity of the patients’ conditions, which could hinder their recovery. Another clinical trial conducted in patients with HIV infection evaluated the effect of CoQ10 administration on CD4+ cell count and markers of liver and kidney function [46]. The study shows that in both groups, control and CoQ10, CD4+ count was significantly increased, without changes in markers of liver and kidney function, although the study does not demonstrate the efficacy of CoQ10 on T cell count, it is worth mentioning that conventional treatment could be responsible for the results obtained, however, in these patients with HIV infection it has been described that their CoQ10 levels endogenous are significantly decreased, so supplementation with CoQ10 helps to restore this deficiency.

In 2012, De Luca et al., examined the effects of antioxidant use, including CoQ10, on clinical benefits, inflammatory response, and antioxidant supplementation in patients with mucocutaneous viral lesions; the study shows that nutraceutical supplementation in patients suffering from viral infections, added to conventional antiviral drug therapy, can substantially improve clinical efficacy and prevent relapses, this same study shows that the use of antioxidants used in the experimental group improves viral load, decreasing it, compared to the control group can decrease viral load [47]. Another study conducted in 2016 evaluated the effect of oral administration of CoQ10 in patients with acute viral myocarditis. A clinical trial was carried out in which the patients were blinded and randomised into three groups based on the treatment: CoQ10; trimetazidine, a cytoprotective anti-ischaemic agent; and combination CQ10 and trimetazidine. Inflammation and oxidative markers and myocardial enzymes were evaluated. These markers decreased in all group, but the combination group showed the most powerful effect. This study demonstrated that the use of CoQ10 in combination with other agents with anti-inflammatory potential could be a therapeutic strategy in infectious diseases [48]. Moreover, a case report evidenced the possible beneficial effect of CoQ10 in anti-N-methyl-D-aspartate receptor (NMDAR) encephalitis, observing improvement in the patient after the use of CoQ10 without response to standard immunotherapy. Although there have not been other studies evaluating the efficacy of CoQ10 supplementation in treating this condition, this report shows a possible complementary therapeutic option [49].

## 6. Conclusions

Although there are few clinical studies that demonstrate the effectiveness of CoQ10 supplementation in infectious diseases, its immunomodulatory and antioxidant effects in chronic diseases have been widely demonstrated. For this reason, its application in infectious processes is feasible and it can present benefits that contribute to the development of a potential new therapy for ID.

## Figures and Tables

**Figure 1 healthcare-10-00487-f001:**
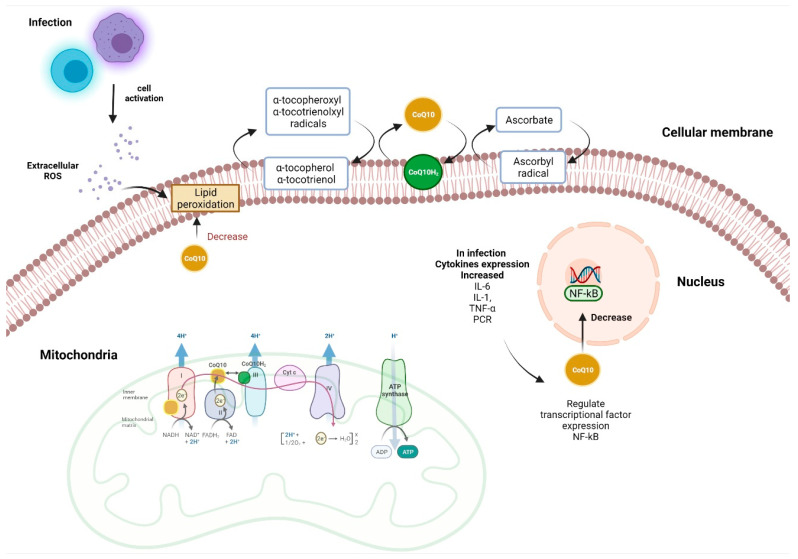
Antioxidant and anti-inflammatory functions of CoQ10. During an infectious process, the cells of the immune response produce cytokines and reactive oxygen species (ROS) as a mechanism to combat a pathogen, these ROS can produce lipoperoxidation, which can be inhibited by the direct antioxidant effect of CoQ10. The main function of CoQ10 in the mitochondria is to transfer electrons to complex III (CIII). By transferring two electrons to Complex III, the reduced form of CoQ10 (ubiquinol) is oxidized to ubiquinone. The ubiquinol pool can be restored by accepting electrons from members of the electron transport chain (CI and CII). The anti-inflammatory effects of CoQ10 may be linked to the regulation of IL-1, IL-6, CRP and TNF-α gene expression through the NF-kB pathway. Figure was created with BioRender software, (Toronto, ON, Canada) ©biorender.com.

**Table 1 healthcare-10-00487-t001:** Antioxidants and anti-inflammatory effects of CoQ10 in infectious diseases: human clinical trials.

Authors	Study Design	Sample Size	Objective	Results
Soltani et al. [44]	Randomized, controlled trial. Follow-up: 7 days-oral administration of 200 mg daily of CoQ10	57 Sepsis patients (randomized) 40 Sepsis patients (completed trial)	Evaluate the effects of CoQ10 supplementation in patients with early sepsis to determine its effects on markers of inflammation and oxidative stress, as well as the clinical impact	CoQ10 group, TNF-α and MDA levels decrease CoQ10 administration improves patient survival
Dominno et al. [45]	Randomized, double-blind, pilot trial	38 patients with severe sepsis	To evaluate the effect of parenteral administration of ubiquinol in patients with severe sepsis on markers of inflammation and markers of vascular endothelial injury.	IL-6 levels at 12, 24, 48 and 72 h were higher in patients who received ubiquinol compared to those who received placebo In the levels of cytochrome C there was also no difference between the groups after the administration of CoQ10 There was no difference in hospital mortality between patients who received ubiquinol versus placebo
Yousefi et al. [46]	Randomized, double blind, placebo controlled, parallel group clinical trial. Follow-up: 3 months-oral administration of 200 mg daily of CoQ10	73 patients with HIV infected	To determine the effect on the effects of CoQ10 on CD4 count in HIV infected patients.	In both study groups the mean TCD4+ cell count increased significantly at the end of treatment. there were no changes in markers of liver and kidney function
De Luca et al. [47]	Two clinical trials were carried out: Double-blind placebo-controlled randomized clinical trial on adult patients with human papillomavirus (HPV) (verruca vulgaris or plantaris) Follow-up: 180 days of nutraceutical or placebo administration Open-label clinical trial on patients affected with herpes disease relapses Follow-up: 90 days of nutraceutical or acyclovir administration	68 patients with relapsing HPV skin warts treated with cryotherapy 89 patients affected with herpes disease relapses	To assess the effect of nutritional therapies with coenzyme Q10, RRR-a-tocopherol, selenium aspartate, and L-methionine associated with established therapies in patients with chronic mucocutaneous infections.	In both trials, the nutraceutical induced significantly faster healing with reduced incidence of relapses as compared to control groups. The group receiving nutraceutical therapy increases GSH levels at the end of the study The combination of nutraceuticals with conventional treatment with acyclovir improved the results with a very pronounced decrease in blood viral load compared to the control treatment group Increased antiviral cytokine and peroxynitrite plasma levels. Plasma antioxidant capacity was higher in the experimental versus control groups.
Shao et al. [48]	Randomized, double blind. Three study groups: CQ10 group, trimetazidine group and CQ10 + trimetazidine group Follow-up: two weeks in patients with acute viral myocarditis	CQ10 group (*n* = 42), trimetazidine group (*n* = 39), and CQ10 + trimetazidine group (*n* = 43)	To evaluate the effect of CQ10 and trimetazidine as monotherapy and in combination for the treatment of acute viral myocarditis.	Inflammation and oxidative markers and myocardial enzymes decreased in all group, but the combination group showed the most powerful effect
Rangel-Guerra et al. [49]	Case report	Case report (1)	Report the case of a patient with anti-NMDAR encephalitis than after poor response to standard immunotherapy, and demonstrate improvement after starting coenzyme Q10 (CoQ10) supplementation	A significant improvement is observed in the patient after the administration of CoQ10, however the results are not conclusive, the effects could be due to the late response to immunotherapy

## Data Availability

Not applicable.

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
