# Peer review of "Antioxidant and Anti-Inflammatory Effects of Coenzyme Q10 Supplementation on Infectious Diseases"

_healthcare, 2022, doi:10.3390/healthcare10030487_

Round 1

Reviewer 1 Report

The review of Sifuentes-Franco et al aims to summarise the clinical evidence that shows the protective role of CoQ10 supplementation  against inflammation and oxidative stress caused by infectious diseases. 

Points that could be improved : 

Point 1. Long phrases and some words are not very correctly used which makes  the text  difficult to read in many paragraphs.   

for example in  Lines 67-71 : rephrase as "There are several processes in which OS plays a fundamental part, including as aging, neuro- 69 degenerative illnesses, diabetes, and autoimmune disorders, where a decreasing level of CoQ10 is evident, and supplementation with CoQ10 has considerable health advantages" rather than " Supplementation with CoQ10 has great health benefits, as there are many processes, such as aging, neurodegenerative diseases, diabetes and autoimmune diseases, in which OS plays a funda- mental role, where a decreased level of CoQ10 is seen"...

Line 147 use "As a result, it's reasonable to believe that ..."  rather than.  "It is consistent to think ..." 

Line 157 :" There are few studies that have evaluated the effect of CoQ10 supplementation in 157 infectious diseases (table 1)" rephrase as "A number of studies ......"

An native English speaker, to a grammar checker should have more suggestions ...

Point 2. Data source should be described in more detail or omitted.  it is quiet general. please refer to keywords used, no of refs retrieved. the bibliography of the subject is not very extensive. the authors thought failed to cite quite a few  :

Shao, Liang, et al. "Combination therapy with coenzyme Q10 and trimetazidine in patients with acute viral myocarditis." Journal of Cardiovascular Pharmacology 68.2 (2016): 150-154.

Rangel-Guerra, R., Camara-Lemarroy, C.R., Garcia-Arellano, G. et al. Could coenzyme Q10 supplementation have a role in the treatment of anti-NMDA receptor encephalitis?. Acta Neurol Belg 115, 85–86 (2015). https://doi.org/10.1007/s13760-014-0299-6

Alexander, Jan, et al. "Early nutritional interventions with zinc, selenium and vitamin D for raising anti-viral resistance against progressive COVID-19." Nutrients 12.8 (20

Liu, HT., Huang, YC., Cheng, SB. et al. Effects of coenzyme Q10 supplementation on antioxidant capacity and inflammation in hepatocellular carcinoma patients after surgery: a randomized, placebo-controlled trial. Nutr J 15, 85 (2015). https://doi.org/10.1186/s12937-016-0205-6

Morrison, Justin T., et al. "Effect of rosuvastatin on plasma coenzyme Q10 in HIV-infected individuals on antiretroviral therapy." HIV clinical trials 17.4 (2016): 140-146.

Thomas Hanscheid, Martin P. Grobusch,
Coenzyme Q10 and cerebral malaria in mice: Questionable interpretations, improbable usefulness in humans, Parasitology International, Volume 74, 2020, 101969,
https://doi.org/10.1016/j.parint.2019.101969

Point 3. In Fig 1.  Fonds could be larger. the nulceus scheme is not very accurate and not clearly linked to the text. what are the small and larger blue dots ??? what is then meaning of the arrow towards the inside of the nucleus???

the title next to the  respiratory chain  does not add anything  and is to accurate. it should be removed. a more precise title of the scheme shown and/or a  mitochondrion scheme should be included. the scheme is misleading as  its is shown in large scale in the cytoplasm while it is  part of mitochondria ( which are  not shown )

Point 4  : abbreviations such as "ID" or "OS" should be omitted.  There aren't many repetitions in the text, which makes the abbreviations unnecessary.

Author Response

We appreciate all the comments and suggestions from the reviewers, as this strengthens the information in the manuscript. Each of the recommendations have been considered, and changes have been marked in blue within the second version of the document and are discussed below:

  1. We have made the suggested changes and have sent the manuscript for review by an English language editing service
  2. We have included the suggested articles in the revised manuscript. The articles by Shao et al. and Rangel-Guerra, in which the authors explore the anti-inflammatory and antioxidant effect in infectious diseases, have been included in the ‘Anti-inflammatory and immune functions’ section. The rest of the suggested articles, as they do not have a clinical impact in the evaluation of the effect of the administration of CoQ10, are included in the rest of the manuscript.
  3. We have improved this figure based on the reviewer’s comments.
  4. We have not removed the abbreviations ‘ID’ and ‘OS’ because they each appear more than three times in the text

Reviewer 2 Report

Section 3, lines 56-58: in my opinion, the role of CoQ10 in OXPHOS should come first.

Line 64: CoQ10 has more relation to quinones, and not to vitamins.

Section 4.

Before referring to antioxidant functions, the authors must describe where oxidative stressors originate.

The description of the biochemistry of CoQ10 in lines 77-78 is incorrect.

Lines 87-88: please specify how ubiquinone and ubiquinol are related and which factors participate.

Line 91: the reference to alpha-tocopherol is not informative enough. There are many more relations.

Lines 102-104 are quite inconclusive. This physology aspect is not described sufficiently.

Lines 117-119: the relation to oxidative stress in cancer is speculative. When talking about cancer, the metabolic characteristics of each tumor need to be described.

Lines 157: the authors must approach the topic of CoQ10 administration in a better way. What is the recommended daily dose? Which blood levels should be achieved?

The situation of sepsis is a complex one that cannot be explained by oxidative changes alone. The authors must describe the multiorgan involvement of each case precisely. Biomarkers are not valid end-point parameter.

Author Response

We appreciate all the comments and suggestions from the reviewers, as this strengthens the information in the manuscript. Each of the recommendations have been considered, and changes have been marked in blue within the second version of the document and are discussed below:

Point 1. Section 3, lines 56-58: in my opinion, the role of CoQ10 in OXPHOS should come first.

Response: We have rearranged the information in section 3 concerning the main functions of CoQ10.

Point 2. Line 64: CoQ10 has more relation to quinones, and not to vitamins.

Response: We have improved the wording to address this comment, highlighting that the similarity with vitamins is in terms of their biological functions.

Point 3. Section 4: Before referring to antioxidant functions, the authors must describe where oxidative stressors originate

Response: We have provided information on ROS generation during infectious diseases: ‘During the infectious process, there could be an imbalance between oxidative and antioxidant species. As part of the body’s response to attack pathogens, the expression of myeloperoxidase, NADPH oxidase and nitric oxide synthase (iNOS) is induced in phagocytic cells, leading to increased production of ROS and RNS. In infection there is abundant iNOS activity in macrophages and leukocytes leading to increased production of nitric oxide (NO), which in the presence of superoxide reacts to form highly toxic peroxynitrite. The oxidants generated activate the redox-sensitive NF-κB signalling pathway, which promotes the expression of proinflammatory cytokines, chemokines and cell adhesion receptors, all of which are involved in radical production and persistence of inflammation.’

Point 3. The description of the biochemistry of CoQ10 in lines 77-78 is incorrect.

Response: We have modified the information on lines 77–78: ‘CoQ10 is in equilibrium between a reduced form, ubiquinol, after receiving two electrons, and an oxidised form, ubiquinone. In the respiratory chain, this redox cycle occurs via a two-step transfer (one electron per step), producing the semiquinone intermediate.’

Point 4: Lines 87-88: please specify how ubiquinone and ubiquinol are related and which factors participate.

Response: We have added the following information to address this comment: ‘CoQ10 is reduced to CoQ10H2 with electrons provided by other redox reactions in the mitochondria, namely by mitochondrial glycerol 3-phosphate dehydrogenase, mitochondrial dihydroorotate dehydrogenase and electron-carrying flavoprotein dehydrogenases (ETFDHs).’

Point 5: Line 91: the reference to alpha-tocopherol is not informative enough. There are many more relations.

Response: We have added information regarding the interaction between ubiquinol and alpha-tocopherol: ‘Reducing compounds such as ubiquinol can regenerate alpha-tocopherol from the tocopherol radical. This recycling is dependent on the activity of mitochondrial succinate dehydrogenase, which provides the reducing equivalents for the conversion of ubiquinone to ubiquinol. Microsomal NADPH-dependent reduction of the phenoxyl radical in vitamin E achieves maximum efficiency in the presence of CoQ10.’

Point 6: Lines 102-104 are quite inconclusive. This physology aspect is not described sufficiently.

Response: We have added the following information to the manuscript: ‘Lipoperoxides present in pathogens and oxidising agents generated during the infectious process induce the signalling pathway in monocytes that activates NF-kB, resulting in the release of TNF-alpha. CoQ10 can reduce the activation of this signalling pathway due to its radical scavenging activity.

Point 7: Lines 117-119: the relation to oxidative stress in cancer is speculative. When talking about cancer, the metabolic characteristics of each tumor need to be described.

Response: We have modified the wording as follows: ‘Another pathology related to oxidative stress and inflammation is cancer. Although there are few studies on CoQ10 supplementation as an adjuvant in cancer treatment, there is evidence of remission after supplementation with this compound. In addition, CoQ10 has demonstrated anti-inflammatory effects and decreased oxidative stress in patients with hepatocellular carcinoma.’

Point 8: Lines 157: the authors must approach the topic of CoQ10 administration in a better way. What is the recommended daily dose? Which blood levels should be achieved?

Response: We have added information to section 3 to address this comment: ‘There are no established minimum or maximum effective doses; consumption of 200 mg twice a day is required to achieve a therapeutic blood level of > 2.5 µg/mL, although the doses used have varied according to the type of disease. Regarding its modulatory effect on inflammation, supplementation with 100-300 mg/day for 2-3 months reduced pro-inflammatory parameters.’

Point 9. The situation of sepsis is a complex one that cannot be explained by oxidative changes alone. The authors must describe the multiorgan involvement of each case precisely. Biomarkers are not valid end-point parameter.

Response: ‘It is correct that oxidative changes cannot define the outcome of an infectious process, as tissue-specific effects depend on the pathogen and the site of infection, but oxidative stress is likely to be present in most circumstances. The aim of the manuscript is to address the effect of CoQ10 on oxidative stress and inflammation in infectious diseases, which is why not all the explanatory parameters of the evolution of each disease have been explored in depth.’

Reviewer 3 Report

In this paper, the authors make a comprehensive summary of the literature regarding coenzyme Q10 supplementation on infectious diseases. Although this works provides a good compilation of information, several minor modifications should be made in order to make this paper publishable.

  1. In my opinion (but just a personal opinion), first sentence of section 3 (lines 55-56) fits better at the beginning of the Introduction section.
  2. NQO1 abbreviation (line 90) lacks the description (NAD(P)H dehydrogenase [quinone] 1, I think) and should be placed to make the text easier to follow. Same happens with CRP (line 102) and C-reactive protein.
  3. In line 169, I think that the corresponding reference is 37 instead of 38. Reference 38, however, should be placed in paragraph enclosing lines 176-182 (maybe after De Luca et al)
  4. As another personal suggestion as in Question 1, I think that placing the Data source section after the Conclusion section would make the paper easier to follow.
  5. Acknowledgements section (line 200-202) has not been written. I encourage you to provide information about the different founding sources supporting the authors of this review.

All these topics considered, just congratulate you for this paper.

Author Response

We appreciate all the comments and suggestions from the reviewers, as this strengthens the information in the manuscript. Each of the recommendations have been considered, and changes have been marked in blue within the second version of the document and are discussed below:

1. In my opinion (but just a personal opinion), first sentence of section 3 (lines 55-56) fits better at the beginning of the Introduction section.

Response: We have incorporated the recommendations suggested to the manuscript format.

2. NQO1 abbreviation (line 90) lacks the description (NAD(P)H dehydrogenase [quinone] 1, I think) and should be placed to make the text easier to follow. Same happens with CRP (line 102) and C-reactive protein.

Response: We have incorporated the recommendations suggested to the manuscript format.

3. In line 169, I think that the corresponding reference is 37 instead of 38. Reference 38, however, should be placed in paragraph enclosing lines 176-182 (maybe after De Luca et al)

Response: We appreciate the observation. We have incorporated the recommendations suggested to the manuscript format.

4. As another personal suggestion as in Question 1, I think that placing the Data source section after the Conclusion section would make the paper easier to follow.

Response: We have incorporated the recommendations suggested to the manuscript. Changes have been marked in blue within the second version of the document.

5. Acknowledgements section (line 200-202) has not been written. I encourage you to provide information about the different founding sources supporting the authors of this review.

Response: We have removed the acknowledgments section because it does not apply to this study.

Reviewer 4 Report

  1. There are some grammatical, alignment and typographical errors are noted in the manuscript and it should be thoroughly checked and corrected throughout the manuscript. For example, in line number 21, the word “treatment” may be as “the treatment”; in line number 116, “a placebo” as “placebo”; in line number 189, “that it” as “it”.

  1. Check the abbreviations throughout the manuscript and introduce the abbreviation when the full word appears the first time in the text and then use only the abbreviation (For example, infectious diseases (ID), CRP, IL-6, TNF-α, nuclear factor kappa B (NF-κB), etc.,). And it should be in both abstract as well as in the remaining part of the manuscript. Make a word abbreviated in the article that is repeated at least three times in the text, not all words need to be abbreviated.

  1. The introduction part appears less informative about infectious diseases and the treatment regimens available, thus this section should be indicated as detailed to understand the manuscript in clear.

  1. The literature search should be described in detail. Hence, the authors are encouraged to include keywords used along with the database or search engine etc., which may be included under the heading “Data source” since the authors focused on the literature review also. How many articles were obtained from each of the search engines? What is the inclusion and exclusion criteria? What is the type of article included in this manuscript? Instead of mentioning up to December 2021.

  1. In subheading 3. Coenzyme Q10, the authors are encouraged to include more data about the natural sources since it has wide medicinal activity.

  1. In table 1, the initial cited with the authors name should be removed and it should be as per the journal instruction.

  1. The table and figure legends should be improved and a proper footnote should be given. All legends should have enough description for a reader to understand the table and figure without having to refer back o the main text of the manuscript. For example, the necessary expansion may be given for abbreviations used.

  1. The authors may include toxicity study or safety measures data, if available for better understanding.

Author Response

We appreciate all the comments and suggestions from the reviewers. We have considered each of the recommendations and have made changes to the manuscript, which we have marked in blue. We believe that these changes have strengthened the manuscript. Below we provide a point-by-point response to the reviewers.

1. There are some grammatical, alignment and typographical errors are noted in the manuscript and it should be thoroughly checked and corrected throughout the manuscript. For example, in line number 21, the word “treatment” may be as “the treatment”; in line number 116, “a placebo” as “placebo”; in line number 189, “that it” as “it”.

Response: We considered the recommendation made by the reviewer

2. Check the abbreviations throughout the manuscript and introduce the abbreviation when the full word appears the first time in the text and then use only the abbreviation (For example, infectious diseases (ID), CRP, IL-6, TNF-α, nuclear factor kappa B (NF-κB), etc.,). And it should be in both abstract as well as in the remaining part of the manuscript. Make a word abbreviated in the article that is repeated at least three times in the text, not all words need to be abbreviated.

Response: We have incorporated the recommendations suggested to the manuscript

3. The introduction part appears less informative about infectious diseases and the treatment regimens available, thus this section should be indicated as detailed to understand the manuscript in clear.

Response: We have incorporated the recommendations suggested to the manuscript

4. The literature search should be described in detail. Hence, the authors are encouraged to include keywords used along with the database or search engine etc., which may be included under the heading “Data source” since the authors focused on the literature review also. How many articles were obtained from each of the search engines? What is the inclusion and exclusion criteria? What is the type of article included in this manuscript? Instead of mentioning up to December 2021.

Response: We have complemented the data source section

5. In subheading 3. Coenzyme Q10, the authors are encouraged to include more data about the natural sources since it has wide medicinal activity.

Response: We have incorporated the recommendations suggested to the manuscript. Changes have been marked in blue within the second version of the document

6. In table 1, the initial cited with the authors name should be removed and it should be as per the journal instruction.

Response: We appreciate the observation. We have incorporated the recommendations suggested to the manuscript format

7. The table and figure legends should be improved and a proper footnote should be given. All legends should have enough description for a reader to understand the table and figure without having to refer back o the main text of the manuscript. For example, the necessary expansion may be given for abbreviations used.

Response: We considered the recommendation made by the reviewer

8. The authors may include toxicity study or safety measures data, if available for better understanding.

Response: We appreciate the observation and incorporated the recommendations suggested to the manuscript format

Round 2

Reviewer 1 Report

The  manuscript has been improved considerably

Reviewer 2 Report

The authors have improved the manuscript.

The formatting of citations is not correct.

The formatting of tables is not correct.